# Búsqueda de Vecindad Variable para el *Board Packing Problem*

**Sergio Pérez-Peló**\*
Department of Computer Science
Universidad Rey Juan Carlos
C/Tulipan, S/N, 28933, Móstoles, Madrid, Spain
`sergio.perez.pelo@urjc.es`

**Jesús Sánchez-Oro**
Department of Computer Science
Universidad Rey Juan Carlos
C/Tulipan, S/N, 28933, Móstoles, Madrid, Spain
`jesus.sanchezoro@urjc.es`

**Anna Martínez-Gavara**
Departament d'Estadística i Investigació Operativa
Universitat de València
C/Doctor Moliner 50, 46100, Burjassot, Valencia, Spain
`anna.martinez-gavara@uv.es`

**Ana D. López-Sánchez**
Universidad Pablo de Olavide
Ctra. de Utrera, km 1, 41013, Sevilla, Sevilla, Spain
`adlopsan@upo.es`

## Abstract

En el *Board Packing Problem* (BoPP) se considera un tablero rectangular dividido en celdas con $n$ filas y $m$ columnas, donde cada celda tiene asignado un valor (positivo o negativo). El problema consiste en asignar un subconjunto de rectángulos de un conjunto dado, cada uno con diferentes costes, sobre diferentes celdas del tablero, de manera que se obtendrá el beneficio de todas las celdas que queden cubiertas. El objetivo del BoPP es colocar estos rectángulos sobre el tablero de manera que se maximice el beneficio total, calculado como la suma de beneficios de las celdas cubiertas menos el coste acumulado de utilizar los rectángulos que componen la solución. Hay que tener en cuenta que los beneficios de una celda sólo se obtienen una vez, es decir, se admiten solapes entre rectángulos, pero el beneficio de las celdas solapadas se recoge una sola vez. En este trabajo, se propone un enfoque basado en Variable Neighborhood Descent (VND) para resolver el BoPP. Se proponen dos procedimientos constructivos para generar la solución inicial de la que partirá el VND: un enfoque totalmente voraz y un método semi-aleatorio para favorecer la diversidad. En la fase de experimentos, se analiza la contribución de cada componente del algoritmo final y, a continuación, se realiza una prueba competitiva para evaluar el rendimiento del algoritmo comparándolo con el mejor

---
\*Use footnote for providing further information about author (webpage, alternative address)—*not* for acknowledging funding agencies.

método encontrado en el estado del arte. La superioridad de la propuesta se apoya en pruebas estadísticas no paramétricas.

## 1. Introducción

La transición de la sociedad hacia las llamadas Ciudades Inteligentes (*Smart Cities*) requiere de un acompañamiento tecnológico que permita llevar a cabo con éxito esta transición. Uno de los pilares fundamentales en esta tarea es la elaboración de algoritmos eficientes que permitan obtener soluciones óptimas (o, al menos, de alta calidad) en tiempos de cómputo reducidos. En este contexto, las heurísticas y las metaheurísticas emergen como una alternativa a tener en cuenta. Por lo tanto, definir problemas de optimización que ayuden a construir productos de software que faciliten esta tarea es una importante responsabilidad de la comunidad científica. Además, es posible que, a partir de la definición de un problema de optimización que modele una situación específica, se puedan definir de forma más sencilla formulaciones de problemas que aborden situaciones similares. En este sentido, los investigadores en [1] propusieron un nuevo problema de optimización, denominado *Board Packing Problem* (BoPP). En este trabajo se demuestra, a su vez, que el problema definido es $\mathcal{NP}$-duro.

En concreto, aportar algoritmos para la resolución del *Board Packing Problem* puede ser beneficioso para ámbitos como la visión artificial. Un ejemplo de aplicación sería trasladar el problema al estudio de la localización de placas solares en un terreno dado. Las restricciones de este problema hacen que el diseño y distribución de las placas sobre un terreno concreto se conviertan en una tarea realmente demandante para operarios humanos. En este aspecto, la tecnología aplicada para resolver el *Board Packing Problem* podría trasladarse de manera que el tiempo dedicado a esta tarea se reduzca enormemente. En concreto, bastaría con modelar el terreno como un espacio rectangular, dividirlo en celdas (una operación clásica en la cartografía) y asociar beneficios positivos a las celdas resultantes en las que convenga colocar una placa y beneficios negativos a las zonas en las que no. Además, al igual que se consideran diferentes tipos de rectángulos con diferentes costes asignados, se pueden considerar diferentes tipos de placas para un mismo terreno.

Nuestro interés por resolver el BoPP viene motivado por una situación realista que se estaba dando en nuestro país. Recientemente, en España, se utilizaron aviones para fotografiar el terreno, con el objetivo de descubrir irregularidades urbanísticas. Debido a ello, en el año 2022 la regularización catastral destapó fraudes en casi dos millones de viviendas. Además de para actualizar el Catastro, las imágenes obtenidas se utilizaron para recopilar información cartográfica, geográfica, forestal y agraria, entre otras. El procedimiento de regularización catastral tiene como objetivo garantizar que la descripción catastral de los bienes inmuebles se ajusta a la realidad y, por tanto, que el Impuesto sobre Bienes Inmuebles de Naturaleza Urbana es abonado correctamente por el propietario del inmueble. Por ejemplo, el pago es diferente en función del terreno (suelo urbano o suelo rústico). A su vez, si nos centramos en el suelo urbano, no es lo mismo que lo ocupe una vivienda, un aparcamiento, un comercio o una nave, entre otros. Para identificar la propiedad, los aviones toman imágenes de algunas parcelas del terreno; en nuestro problema, se pueden definir como celdas en el tablero, aunque el hecho de que los aviones estén fotografiando implica que suponen un coste para los ayuntamientos [2]. Al mismo tiempo, una vez que una parcela se identifica como incorrecta o fraudulenta, el propietario de la parcela debe hacer frente al pago correcto, por lo que el municipio recibe un ingreso [3, 4].

Formalmente, el BoPP considera un tablero rectangular dividido en celdas con $n$ filas y $m$ columnas. En adelante, la celda $(i, j)$ para $i = 1, \ldots, n$ y $j = 1, \ldots, m$ representa la celda situada en la fila $i$ y la columna $j$. Cada celda tiene un beneficio asociado, un valor entero $g_{ij}$ que representa el ingreso obtenido si se coloca un rectángulo en la celda $(i, j)$. Los beneficios pueden ser positivos, representando una ganancia, o negativos, representando una penalización. Dado un número, $R$, de rectángulos, cada rectángulo $r = 1, \ldots, R$ se caracteriza por una altura, $h_r$ y una anchura, $w_r$, y además, tiene un coste asociado, $c_r$, si se utiliza el rectángulo $r$ en la solución. El objetivo del BoPP es colocar un subconjunto de rectángulos sobre las celdas del tablero para maximizar el beneficio, calculado como la suma de los ingresos de las casillas cubiertas por cada rectángulo seleccionado menos el coste de utilizar dichos rectángulos. En el problema considerado se permite la superposición de rectángulos; sin embargo, los ingresos de una celda del tablero sólo pueden recogerse una vez. En términos matemáticos, la función objetivo asociada a una solución $S$ puede definirse de la siguiente manera:

$$f(S) = \left( \sum_{i=1}^{n} \sum_{j=1}^{m} g_{ij} \cdot x_{ij} \right) - \left( \sum_{r=1}^{R} c_r \cdot y_r \right)$$

donde $x_{ij}$ es una variable binaria que toma el valor 1 si la celda $(i, j)$ del tablero está cubierta por, al menos, un rectángulo y 0 en caso contrario; y $y_r$ es una variable binaria que toma el valor 1 si el rectángulo $r$ ha sido asignado en el tablero y 0 en caso contrario. El objetivo es encontrar una solución $S^*$ que maximice los ingresos, es decir, que maximice el valor de $f(S)$. Más formalmente,

$$S^* = \arg\max_{S \in \mathcal{SS}} f(S)$$

donde $\mathcal{SS}$ representa el conjunto de soluciones factibles de BoPP.

En aras de la claridad, el ejemplo incluido en la Figura 1 muestra una instancia que nos ayuda a ilustrar el BoPP. El ejemplo consiste en un tablero con $n = 5$ filas y $m = 5$ columnas. El objetivo es colocar como máximo dos rectángulos, $R = 2$. El primer rectángulo (en azul), $r = 1$, se caracteriza por $h_1 = 4$, $w_1 = 2$ y $c_1 = h_1 \cdot 10 + w_1$. El segundo rectángulo (en rojo), $r = 2$, está definido por $h_2 = 2$, $w_2 = 2$ y $c_2 = h_2 \cdot 10 + w_2$.

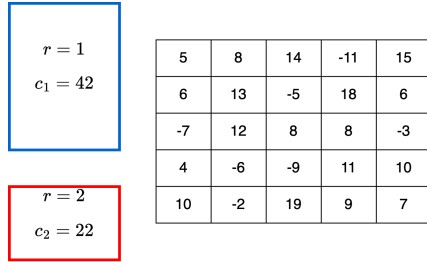

Figura 1: Ejemplo de instancia para el BoPP.

La Figura 2 ilustra tres soluciones factibles diferentes del BoPP para el ejemplo, denotadas como $S_1$, $S_2$, y $S_3$. Sin pérdida de generalidad, a lo largo del resto del manuscrito, la celda en la que se encuentra un rectángulo representa su esquina superior izquierda. Como puede verse en la Figura 2a, el rectángulo 1 se ha situado en la celda $(2, 4)$ y el rectángulo 2 en la celda $(1, 1)$, tomando como referencia la esquina superior izquierda de ambos. Por lo tanto, para $S_1$, el valor de la función objetivo se puede calcular como la suma de los ingresos de las celdas cubiertas por los rectángulos menos la suma de los costes de ambos rectángulos. Por tanto, el valor para esta solución es: $(5+8+6+13+18+6+8-3+11+10+9+7)-(22+42) = 98-64 = 34$. El mismo razonamiento puede seguirse para calcular los valores de la función objetivo obtenidos por la solución $S_2$ (representada en la Figura 2b): $(8+14+13-5-11+15+18+6+8-3+11+10)-(22+42) = 84-64 = 20$. El valor de la función objetivo $S_2$ de la solución se obtiene situando el rectángulo $r = 1$ en la celda $(1, 4)$ y el rectángulo $r = 2$ en la celda $(1, 2)$, tomando de nuevo como referencia la esquina superior izquierda de ambos. Sin embargo, el cálculo de la solución $S_3$ (mostrada en la Figura 2c) es ligeramente diferente, ya que en esta solución hay un solapamiento entre los rectángulos asignados, dado que el rectángulo $r = 1$ se ha colocado en la celda $(2, 3)$ y el rectángulo $r = 2$ en la celda $(2, 2)$. En este caso, los ingresos de las celdas cubiertas por ambos rectángulos sólo se pueden contar una vez. Por lo tanto, el cálculo de la función objetivo sería el siguiente $(13+12-5+8+18+8-9+11+19+9)-(22+42) = 84-64 = 20$. Es importante señalar que, en este problema, no hay diferencia entre una solución con rectángulos superpuestos y una solución que no los contiene. No hay penalización en el valor de la función objetivo debido a la superposición, por lo que $S_2$ y $S_3$ son dos soluciones equivalentes.

Hasta donde sabemos, el BoPP se abordó por primera vez en el trabajo de [1]. Los autores introdujeron formalmente el problema y también demostraron que es $\mathcal{NP}$-duro. Con el fin de resolver el problema, presentan un modelo de programación entera binaria y utilizan CPLEX, un solver comercial, para encontrar soluciones óptimas para un conjunto de instancias de pequeño tamaño, además de obtener límites superiores para instancias de tamaño medio. Además, los autores proponen un algoritmo evolutivo para obtener soluciones heurísticas para aquellas instancias en las que el algoritmo exacto no es capaz de alcanzar soluciones óptimas. En concreto, el algoritmo evolutivo crea una población de soluciones iniciales y, a continuación, se aplican operadores de selección, combinación y mejora para

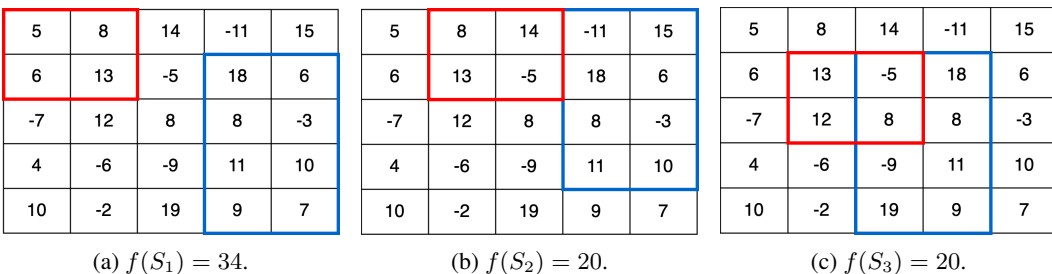

(a) $f(S_1) = 34$.        (b) $f(S_2) = 20$.        (c) $f(S_3) = 20$.

Figura 2: Tres soluciones factibles para la instancia mostrada en la Figura 1.

alcanzar soluciones de alta calidad. Incluyen una amplia experimentación computacional y llegan a la conclusión de que la dificultad del BoPP depende del tamaño y la topografía del tablero, así como del número y la variedad de los rectángulos que pueden asignarse. En este artículo previo, los autores también ofrecen una revisión exhaustiva de los modelos existentes estrechamente relacionados con el BoPP, destacando y comparando sus características clave.

El resto del trabajo se organiza de la siguiente manera: una vez descrito y motivado el problema en la Introducción, en la Sección 2 se dan todos los detalles de la propuesta algorítmica para resolver el BoPP. En la Sección 3 se presentan los resultados computacionales realizados para comprobar la calidad de nuestra metaheurística, incluyendo una comparativa con el algoritmo del estado del arte. Finalmente, la Sección 4 concluye el artículo y discute futuros trabajos.

## 2. Propuesta Algorítmica

En este trabajo se propone un algoritmo basado en la metaheurística Variable Neighborhood Descent (VND). Variable Neighborhood Descent [5] es una variante de la metaheurística Variable Neighborhood Search (VNS) [6, 7], una metodología que trata de escapar de óptimos locales realizando cambios sistemáticos de vecindad, y que habitualmente es utilizada como mecanismo de búsqueda local. En concreto, la metodología parte de una solución inicial y explora sistemáticamente diferentes vecindades. Cada una de ellas será explorada hasta que se encuentre un óptimo local. Si durante la búsqueda se encuentra una mejora, ésta se reiniciará partiendo de la nueva solución encontrada, recorriendo de nuevo la primera vecindad definida. Si, por el contrario, no se encuentra ninguna mejora, se explorará la siguiente vecindad. Este proceso se repite hasta que se alcanza una solución a partir de la cual no se encuentra ninguna mejora recorriendo todas las vecindades definidas. En el contexto del BoPP, pueden considerarse varios vecindarios para explorar el espacio de búsqueda, por lo que VND surge como una metaheurística adecuada para abordar este problema. Remitimos al lector a [8] para un estudio reciente sobre VND.

Existen distintas variantes de la metodología VNS, estando algunas más centradas en la diversificación de la búsqueda y otras en su intensificación [9, 10]. En concreto, la variante VND se centra en este último aspecto. El algoritmo VND se ha utilizado con éxito para resolver varios problemas de optimización combinatoria [11, 12]. En la versión básica de la metodología VNS, se distinguen tres fases diferentes: en la fase de *shake* o perturbación, se modifica de alguna forma una solución inicial contenida en el vecindario bajo exploración para favorecer la diversidad; a continuación, la fase de mejora o *improvement* se encarga de encontrar un nuevo óptimo local aplicando algún algoritmo de mejora, como por ejemplo un método de búsqueda local; por último, en la fase de intercambio de vecindario o *neighborhood change* se decide si reiniciar la búsqueda desde el primer vecindario explorado o explorar el siguiente vecindario definido, dependiendo de si se encuentra o no una mejora de la solución.

### 2.1. Fase de construcción

Como se ha mencionado anteriormente, para aplicar VND es necesario partir de una solución inicial. Como se describe en la última revisión de la metodología [5], la solución inicial para VND puede generarse de diferentes maneras. Las más extendidas son la construcción aleatoria y construcciones que utilizan un procedimiento constructivo más elaborado; incluso, podría utilizarse una metaheurística completa para generar la solución inicial [13].

En este trabajo, proponemos el uso de un procedimiento constructivo inspirado en la metodología Greedy Randomized Adaptive Search Procedure (GRASP) para la generación de la solución inicial. La metaheurística GRASP fue propuesta originalmente en [14] y formalizada en [15, 16]. Se trata de un procedimiento multiarranque, en el que cada arranque corresponde a una iteración del algoritmo. Cada una de estas iteraciones tiene dos fases diferenciadas: la fase de construcción y la fase de mejora. La primera genera una solución factible de alta calidad, mientras que la segunda se encarga de encontrar un óptimo local respecto a una vecindad predefinida, partiendo de la solución obtenida en el paso anterior. Referimos al lector al trabajo original para un mayor detalle en la definición de estas dos fases.

Para resolver el BoPP, se utiliza una modificación de la fase de construcción GRASP para lograr una solución inicial de alta calidad. En concreto, hay que tener en cuenta tres decisiones principales: qué rectángulos colocar, dónde colocarlos y cuántos colocar. La última decisión viene dada por una condición de parada: se añadirán rectángulos a la solución siempre que haya rectángulos disponibles, mientras el valor de la función objetivo no disminuya. Para tomar las dos decisiones restantes, se aplicarán diferentes estrategias de construcción que serán evaluadas experimentalmente. Estas estrategias tienen en cuenta tres elementos fundamentales del problema: el coste de incluir un rectángulo en la solución, el beneficio de hacerlo y las celdas sobre las que se puede colocar un rectángulo dado. En base a estos elementos, se definen dos criterios voraces en los que basaremos nuestros métodos constructivos: el coste de localizar un rectángulo en el tablero y el ratio coste / beneficio asociado a un rectángulo, considerando como su beneficio aquel que obtendría si su esquina superior izquierda[2] se localizase en una posición determinada.

En la Tabla 1 se muestran las cuatro principales estrategias propuestas para generar la solución inicial del BoPP, clasificadas tanto por el criterio utilizado para seleccionar el siguiente rectángulo (Selección de rectángulo) como por el utilizado para seleccionar su posición (Selección de ubicación). Existen dos estrategias principales que se pueden aplicar para cada criterio: *greedy*, que indica que la decisión se realiza de forma completamente voraz (es decir, se selecciona el siguiente mejor elemento), y semisemi-*greedy*, que selecciona el siguiente elemento siguiendo el enfoque inspirado en la fase constructiva GRASP antes mencionada.

| Estrategia | Selección de rectángulo | Selección de ubicación |
|---|---|---|
| SS | Semi-greedy | Semi-greedy |
| GS | Greedy | Semi-greedy |
| SG | Semi-greedy | Greedy |
| GG | Greedy | Greedy |

Tabla 1: Resumen de las cuatro estrategias constructivas propuestas.

En todas las implementaciones de los algoritmos presentados en este trabajo, la primera decisión a tomar siempre será qué rectángulo incluir en la solución y, a continuación, en qué celda se colocará. La estrategia (GG) es la más sencilla de implementar: para la selección voraz de rectángulos, basta con ordenar los rectángulos en función del criterio voraz seleccionado (ratio coste/beneficio o únicamente coste) y, a continuación, para la selección voraz de la ubicación, basta con colocar sobre la celda que aporte el mayor beneficio al rectángulo seleccionado. A continuación, es necesario iterar repitiendo la misma operación hasta que no haya más rectángulos que añadir que no disminuyan el valor de la función objetivo. Cuando se aplica una estrategia semi-*greedy*, se sigue un procedimiento más elaborado. En primer lugar, se construirá una lista de candidatos restringida (RCL, *Restricted Candidate List*) en base a un parámetro $\alpha$, que determinará la voracidad / aleatoriedad del algoritmo (cuanto menor es el valor de $\alpha$, más voraz será su comportamiento, y cuanto mayor sea este valor, más aleatorizado será). Esta RCL contendrá los rectángulos candidatos a formar parte de la solución. A continuación, se selecciona un candidato al azar de la RCL y se procede a tomar la decisión sobre dónde será ubicado. Para ello, en función de la estrategia seguida, se seleccionará la mejor celda en función del beneficio que reportaría colocar la esquina superior izquierda del rectángulo sobre ella (estrategia voraz pura) o se replicará el proceso seguido para seleccionar un rectángulo: se construirá una nueva RCL donde los candidatos pasarán a ser las celdas en las que se puede colocar el

---

[2]La idea de identificar la posición de un rectángulo en función de la celda que ocupa su esquina superior izquierda se aplica con éxito en [1], por lo que parece apropiado continuar con esta nomenclatura.

rectángulo seleccionado. A continuación, se seleccionará una de estas celdas al azar, y se posicionará el rectángulo sobre ella.

## 2.2. *Variable Neighborhood Descent*

La solución inicial generada en la Sección 2.1 no es necesariamente un óptimo local con respecto a ninguna vecindad. Por lo tanto, puede mejorarse con un simple mecanismo de búsqueda local o con un procedimiento más elaborado. En el contexto del BoPP, la fase de mejora consistirá en la aplicación de un algoritmo VND completo. En la metodología VND, los cambios de vecindad se realizan definiendo diferentes vecindades y explorando cada una de ellas de forma determinista. Para definir una vecindad primero es necesario definir un movimiento, que expresará las modificaciones que se realizarán sobre la solución y, por tanto, las soluciones que compondrán la vecindad. En este trabajo, los cuatro movimientos propuestos se basan en la misma idea: mover un rectángulo dado en las cuatro direcciones del espacio alrededor del tablero, intentando encontrar una mejor configuración del mismo. Podemos definir estos movimientos de la siguiente manera: dada una solución $S$ compuesta por un conjunto de rectángulos, y una función $P : \mathbb{Z} \to \mathbb{Z} \times \mathbb{Z}$ que determina la posición $(i, j)$ de un cierto rectángulo en el tablero, siendo $i$ la fila y $j$ la columna de la celda; el movimiento consiste en modificar la posición de un rectángulo desplazándolo una unidad en una de las direcciones disponibles ($d$) en el tablero: arriba (0), abajo (1), izquierda (2), o derecha (3), siempre que el movimiento sea factible (es decir, que el rectángulo no esté fuera de los límites del tablero). Más formalmente,

$$Move(S, r, d) = \begin{cases} P(r) \leftarrow P_1(r) - 1, P_2(r) & \text{si} \quad d = 0 \\ P(r) \leftarrow P_1(r) + 1, P_2(r) & \text{si} \quad d = 1 \\ P(r) \leftarrow P_1(r), P_2(r) - 1 & \text{si} \quad d = 2 \\ P(r) \leftarrow P_1(r), P_2(r) + 1 & \text{si} \quad d = 3 \end{cases}$$

donde $P_1(r)$ representa la coordenada $x$ de un rectángulo $r$ y $P_2(r)$ la coordenada $y$.

Dadas estas definiciones, el algoritmo VND procederá de la siguiente manera: partiendo de la solución generada por el método de construcción definido en la Sección 2.1, se explorarán las vecindades definidas, comenzando hacia arriba y siguiendo la dirección contraria a las agujas del reloj, es decir, arriba-izquierda-abajo-derecha. Cada una de estas vecindades se explorará siguiendo una estrategia de *first improvement*, lo que significa que cuando un movimiento alcanza una solución mejor en términos del valor de la función objetivo, se actualiza la solución, reiniciando la búsqueda. En la metodología VND, siempre que se encuentra una mejora, la búsqueda se reinicia desde la primera vecindad definida. En caso contrario, se seguirá explorando la siguiente vecindad, hasta que no se encuentre ninguna mejora en ninguna de las vecindades exploradas. Hay que tener en cuenta que las vecindades definidas están compuestas por todos los movimientos en la misma dirección que se pueden realizar con todos los rectángulos, evaluando uno a uno todos los rectángulos que forman parte de la solución inicial. Para controlar el número de vecindades bajo exploración, se utiliza el parámetro $k_{max}$. De esta manera, el algoritmo VND iterará desde la vecindad $k = 0$ hasta la vecindad $k = k_{max}$ de la manera descrita previamente.

El algoritmo completo, denominado Multistart VND, se define en el pseudocódigo mostrado en el Algoritmo 1.

Como puede observarse, las fases de construcción y mejora local se repiten durante un cierto número de iteraciones, que se fija como parámetro del algoritmo ($\Delta$). Dados los componentes aleatorios del algoritmo, especialmente en la construcción, iterar sobre él múltiples veces permite añadir diversidad a las soluciones obtenidas. Cabe destacar que, una vez se alcanza un óptimo local, cabría la posibilidad de añadir más rectángulos a la solución sin empeorar el valor de la función objetivo. Para evaluar esta posibilidad, se ha realizado una experimentación consistente en aplicar nuevos mecanismos de búsqueda local basados en intercambios, donde se retiran $N$ rectángulos de la solución, con $0 \leq N \leq 3$, y se añaden $M$ rectángulos, con $0 \leq M \leq 3$. Sin embargo, no se han encontrado mejoras sustanciales en las soluciones obtenidas y se ha observado un que el tiempo de cómputo requerido para aplicar estas búsquedas locales aumenta en dos veces el requerido sin aplicarlas, por lo que se descarta su aplicación.

**Algorithm 1** MS-VND($\Delta, k_{max}, \alpha, \varsigma, R$)

---

1: $\delta = 1$
2: $S^\star \leftarrow \emptyset$
3: **while** $\delta \leq \Delta$ **do**
4:     $S \leftarrow ConstructiveMethod(\alpha, \varsigma, R)$
5:     $S' \leftarrow VND(S, k_{max})$
6:     **if** $f(S') > f(S^\star)$ **then**
7:         $S^\star \leftarrow S'$
8:     **end if**
9: **end while**
10: **return** $S^\star$

---

## 3. Resultados

En esta sección se presentan y discuten los resultados computacionales obtenidos para evaluar el rendimiento del algoritmo propuesto para resolver el BoPP. Para ello, hemos resuelto un conjunto de 167 instancias que se pueden encontrar en la página web https://home.himolde.no/hvattum/benchmarks. Este conjunto se corresponde con el conjunto de instancias consideradas por el método del estado del arte [1].

Los algoritmos se han desarrollado utilizando Java 21 (OpenJDK 21.0.4) y las ejecuciones se han lanzado limitando el tamaño del heap de la Máquina Virtual Java (JVM) a 4GB. Los experimentos se han llevado a cabo en una máquina virtual (VM) que se ejecuta en un host que utiliza una CPU AMD EPYC 7643 a 2.3 GHz.

Agradecemos a los autores del artículo que presenta el método del estado del arte [1] su disponibilidad y amabilidad al compartir el código fuente y el conjunto de instancias utilizadas en su propuesta. El código anterior ha sido compilado y ejecutado en la misma máquina que el código desarrollado en este trabajo, en aras de la equidad. Este código ha sido compilado utilizando g++ 10.5.0, y ha sido ejecutado utilizando los valores de los parámetros reportados en el trabajo anterior.

La fase experimental se ha dividido en dos etapas diferentes. En primer lugar, realizamos la experimentación preliminar (Sección 3.1), diseñada para seleccionar la mejor versión de nuestro algoritmo completo; a continuación, realizamos la experimentación final (Sección 3.2), en la que la mejor configuración del algoritmo desarrollado se pone a prueba frente al método del estado del arte. Es importante destacar que los experimentos preliminares se han realizado con un subconjunto representativo de las 167 instancias. Más concretamente, se ha seleccionado un subconjunto de 72 instancias para evitar el sobre-ajuste.

La fase experimental se ha diseñado para realizar una comparación equitativa. Para ello, se ha seguido la metodología utilizada por los autores previos: se limita el tiempo de ejecución de la metaheurística a 60 segundos por instancia, se realizan diez ejecuciones completas del algoritmo y se informa del mejor valor encontrado en las diez ejecuciones, el valor medio de la función objetivo, el tiempo medio de cálculo y el tiempo medio necesario para encontrar la mejor solución, la desviación estándar entre las 10 ejecuciones. En el caso de los algoritmos voraces, sólo se lanzará una ejecución del algoritmo y una iteración, ya que por definición un algoritmo codicioso siempre devolverá el mismo resultado para la misma entrada.

Debido al tamaño del conjunto de datos, en este trabajo presentamos las tablas resumen de estos experimentos, en las que informaremos, para cada algoritmo: el valor medio del mejor valor encontrado para todas las instancias, $\overline{F.O}$; el tiempo medio de ejecución, $\overline{T}$(s); el tiempo medio para encontrar el mejor valor de la ejecución, $\overline{TTB}$(s); y el número de veces en que un algoritmo dado ha obtenido la mejor solución del experimento, #Mejores.

### 3.1. Resultados preliminares

Dada la alta combinatoria de experimentos requerida para ajustar las estrategias utilizadas para tomar las dos decisiones pertinentes en el BoPP (qué rectángulos utilizar y en qué celda colocarlos), en este manuscrito no se muestra la experimentación completa para ajustar los parámetros de los métodos constructivos. En concreto, se han ajustado los siguientes parámetros: valor $\alpha$ del algoritmo GRASP

(*RND*, es decir, el valor de $\alpha$ toma un valor aleatorio en el rango $(0, 1]$ en cada iteración del algoritmo); y el número de posiciones diferentes ($\varsigma$) en las que se trata de colocar un rectángulo sin perjuicio del valor de la función objetivo antes de descartarlo para formar parte de la solución. Con esos valores prefijados, se realizó la experimentación para decidir qué estrategia (semi-*greedy* o completamente voraz) es mejor para tomar cada una de las decisiones pertinentes en el BoPP. El resultado de esta comparación puede verse en la Tabla 2. Recordemos que las estrategias evaluadas son: elección voraz tanto del rectángulo como de su posición (GG); elección semi-*greedy* tanto del rectángulo como de su posición (SS); elección semi-*greedy* del rectángulo y elección voraz de su posición (SG); y, finalmente, elección voraz del rectángulo y elección semi-*greedy* de su posición (GS). Como puede observarse, la opción que emerge como la mejor, tanto en valor medio de la función objetivo como en número de veces que se obtiene la mejor solución del experimento, es la que utiliza una estrategia semi-*greedy* para la toma de ambas decisiones, con un valor $\alpha$ de 0.50 y un valor $\varsigma$ de 20.

| Estrategia | $\alpha$ | $\varsigma$ | $\overline{F.O}$ | $\overline{T}(s)$ | $\overline{TTB}(s)$ | #Mejores |
|---|---|---|---|---|---|---|
| GG | - | - | 43850.00 | **0.37** | **0.37** | 1 |
| SS | *RND* | 20 | **88261.17** | 3.63 | 1.83 | **32** |
| SG | *RND* | 25 | 72678.57 | 4.31 | 1.99 | 27 |
| GS | *RND* | 25 | 85903.51 | 4.02 | 1.97 | 16 |

Tabla 2: Tabla comparativa de la mejor versión de cada algoritmo. Los mejores resultados están marcados en negrita. Es importante señalar que la versión GG no requiere de los parámetros $\alpha$ ni $\varsigma$, ya que este enfoque es completamente voraz.

A pesar de ello, es necesario evaluar el comportamiento de los algoritmos constructivos cuando se acoplan a la fase de optimización local, ya que este comportamiento puede verse afectado por la vecindad bajo exploración. Así, partir de una solución que *a priori* es peor, puede permitir alcanzar mejores óptimos locales en la vecindad explorada. La Tabla 3 muestra los resultados de la comparación entre las mejores configuraciones de los algoritmos propuestos cuando se incluyen en el algoritmo VND completo.

| Algoritmo | $\overline{F.O}$ | $\overline{T}(s)$ | $\overline{TTB}(s)$ | #Mejores |
|---|---|---|---|---|
| VND-GG | 96975.88 | 11.69 | **0.36** | 0 |
| VND-SS | 101884.06 | **4.19** | 2.23 | 30 |
| VND-SG | 100695.31 | 4.69 | 2.21 | 16 |
| VND-GS | **102007.04** | 4.27 | 2.13 | **38** |

Tabla 3: Tabla comparativa de la mejor versión de VND. Los mejores resultados están marcados en negrita.

Como se puede ver, la situación anterior ha cambiado. Al ejecutar el algoritmo VND completo, observamos que la construcción con mejor rendimiento es la que utiliza una estrategia voraz para seleccionar los rectángulos y una estrategia semi-*greedy* para la ubicación de las celdas (VND-GS), probando en, como máximo, 25 posiciones del tablero. En concreto, obtiene el valor para la función objetivo promedio más alto, obteniendo el mejor valor de la función objetivo en 38 ocasiones. También es interesante observar cómo el tiempo de computación requerido aumenta cuando se emplea una estrategia voraz para tomar ambas decisiones. Esto se debe a que, aunque se obtienen soluciones de menor calidad con este método, se producen muchas mejoras sobre la solución de partida cuando se aplica el mecanismo de optimización local.

### 3.2. Resultados finales

Una vez obtenida la mejor configuración para nuestro algoritmo, procedemos a compararlo con el mejor algoritmo propuesto en la literatura, que es un algoritmo evolutivo [1], denominado EVO en los experimentos. Nótese que, gracias a la disponibilidad de los autores, el código fuente original ha sido compilado siguiendo sus instrucciones y utilizando los mismos valores para los parámetros de compilación y del algoritmo. Además, ambos algoritmos se han ejecutado bajo las mismas condiciones de hardware. Para ello, se ha utilizado el conjunto completo de instancias. La tabla 4 muestra la comparativa entre los dos algoritmos.

| Algoritmo | $\overline{F.O}$ | $\overline{T}$(s) | $\overline{TTB}$(s) | #Mejores |
|-----------|------------------|-------------------|---------------------|----------|
| VND | **84769.92** | 8.46 | 4.55 | **131** |
| EVO | 83191.81 | **1.09** | **0.66** | 41 |

Tabla 4: Tabla comparativa de la mejor versión de VND con el método propuesto en la literatura (EVO). Los mejores resultados están marcados en negrita.

Como puede observarse, el algoritmo propuesto supera al algoritmo actual en términos de valor de la función objetivo. Sin embargo, el tiempo medio requerido para obtener la mejor solución es 8 veces mayor de media en nuestra propuesta, y el tiempo necesario para encontrar la mejor solución es aproximadamente 7 veces mayor. La conclusión que se puede obtener de estos resultados es que VND es una buena opción cuando se quiere obtener resultados de alta calidad en tiempos razonables, mientras que EVO es una mejor estrategia cuando se busca obtener resultados de manera rápida y es razonable sacrificar calidad de las soluciones en pos de un mayor rendimiento.

Para comprobar la superioridad de la propuesta, se realizó la prueba de Wilcoxon comparando los resultados de los algoritmos. La prueba de Wilcoxon establece como hipótesis nula que no hay diferencia (en términos de tendencia central) entre los dos grupos comparados. La hipótesis alternativa establece que existe una diferencia (en términos de tendencia central) entre los dos grupos. Esta prueba, realizada con el programa estadístico SPSS, arroja un resultado que indica un $p$-valor inferior a 0.001. Por lo tanto, se rechaza la hipótesis nula, lo que indica que no hay pruebas en contra de la afirmación de que existen diferencias significativas entre los dos algoritmos.

## 4. Conclusiones

Este trabajo aborda un problema $\mathcal{NP}$-difícil, el Board Packing Problem (BoPP). Este problema de optimización combinatoria ha sido propuesto recientemente en [1] y el objetivo es maximizar el beneficio obtenido al asignar un conjunto de rectángulos en un tablero rectangular dividido en celdas con $n$ filas y $m$ columnas. Cada rectángulo asignado tiene un coste y cada celda proporciona un beneficio. Por tanto, el beneficio final se mide como los ingresos obtenidos menos el coste pagado.

Para solucionar el BoPP, proponemos aplicar la metaheurística *Variable Neighborhood Descent* (VND), explorando cuatro vecindarios diferentes, y utilizando un mecanismo semi-*greedy* para la construcción de soluciones iniciales. Por un lado, el uso de VND está más que justificado, ya que en este problema tiene sentido explorar más de un vecindario; en nuestro caso, hemos considerado cuatro vecindarios diferentes basados en movimientos de los rectángulos sobre el tablero en las cuatro direcciones del espacio. Por otro lado, se han implementado estrategias inteligentes para construir soluciones iniciales desde cero basadas en dos decisiones: qué rectángulos colocar y dónde colocarlos en el tablero. Además, se han probado dos criterios voraces (relación coste/beneficio del rectángulo y coste del rectángulo).

La experimentación realizada demuestra la superioridad de la propuesta en comparación con la propuesta de [1] sobre el mismo conjunto de instancias. Además, se aportan pruebas estadísticas para afirmar que existen diferencias significativas entre las propuestas y que el algoritmo desarrollado supera al estado del arte.

Como trabajos futuros, sería interesante resolver el BoPP incluyendo nuevas restricciones, más cercanas a escenarios reales, como la ubicación de paneles solares en un área para maximizar la energía producida minimizando el coste de despliegue.

## Agradecimientos

Agradecemos a los autores del trabajo previo [1] su disponibilidad y amabilidad al compartir el código fuente y el conjunto de instancias utilizadas en su propuesta.

Financiación: Esta investigación ha sido parcialmente apoyada por el Ministerio de Ciencia e Innovación de España (Proyecto con Ref. PID2021-125709OA-C22 y PID2022-139543OB-C41), MCIN /AEI / 10. 13039/ 501100011033/ FEDER, y la Unión Europea "NextGenerationEU"/PRTR, la Generalitat Valenciana (Proyecto con Ref. CIAICO / 2021 / 224), y "Ministerio para la Transformación

Digital y de la Función Pública" (Proyecto con Ref. TSI-100930-2023-0003, AI4DDS: Artificial Intelligence for Data Driven Solutions) y de la Comunidad Autónoma de Madrid, con el proyecto CIRMA-CM (referencia TEC-2024/COM-404).

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
