# OpenReview forum: "Búsqueda de Vecindad Variable para el Board Packing Problem"
_MAEB/2025/Congreso — MAEB 2025_

### Official Review · Reviewer_nXv1 · 2025-03-12
**Búsqueda de Vecindad Variable para el Board Packing Problem**

**Rating:** 5
**Confidence:** 2

**Review:**

Este trabajo aborda el Board Packing Problem (BoPP), un problema NP-difícil de optimización combinatoria, cuyo objetivo es maximizar el beneficio al ubicar rectángulos en un tablero dividido en celdas. Para resolverlo, se propone la metaheurística Variable Neighborhood Descent (VND), explorando cuatro vecindarios distintos y empleando un enfoque semi-greedy para la construcción de soluciones iniciales. Se implementan estrategias inteligentes para decidir qué rectángulos colocar y dónde ubicarlos, evaluando dos criterios voraces. Los experimentos demuestran que la propuesta supera a la solución de [1].

El artículo está redactado de manera clara y estructurada, lo que contribuye a una lectura fluida y una comprensión de los conceptos presentados. El tema abordado es relevante dentro del campo de la optimización combinatoria. A lo largo del texto, se describe de manera detallada un algoritmo que se posiciona como una alternativa competitiva frente a las soluciones existentes en la actualidad. Su rendimiento y eficiencia han sido evaluados, demostrando que es capaz de ofrecer resultados comparables a los de los mejores algoritmos existentes en la literatura.

Sugiero a los autores que revisen el escrito porque hay algunos "and" y algún pie de figura está en inglés.

---

### Official Review · Reviewer_MSjk · 2025-03-17
**El trabajo aborda la resolución del Board Packing Problem (BoPP) mediante un enfoque multiarranque que integra diversas variantes de la fase constructiva de las soluciones iniciales con un procedimiento de mejora basado en Variable Neighborhood Descent (VND). El documento está bien redactado y presenta resultados de alta calidad, demostrando un desempeño competitivo en comparación con el estado del arte. Se observan algunos errores menores que deben ser corregidos para su aceptación final.**

**Rating:** 5
**Confidence:** 5

**Review:**

El trabajo aborda la resolución del Board Packing Problem (BoPP) mediante un enfoque multiarranque que integra diversas variantes de la fase constructiva de las soluciones iniciales con un procedimiento de mejora basado en Variable Neighborhood Descent (VND). El documento está bien redactado y presenta resultados de alta calidad, demostrando un desempeño competitivo en comparación con el estado del arte. Se observan los siguientes errores menores que deben ser corregidos en su versión final:

1) En el ejemplo de la Sección 1, concretamente, en las líneas 84 a 97, la solución referida como S2, corresponde con la S3 de la Figura 2 y la S3 con la S2. Lo más sencillo sería intercambiarlas en la Figura.

2) En las líneas 127 y 128 de la propuesta algorítmica, se hace referencia a VND como una metaheurística. Sin embargo, VND es un procedimiento de mejora local, que no combina las fases de exploración y explotación que forman parte de una metaheurística. Se podría escribir que VND surge como un procedimiento de mejora que integra los diferentes movimientos considerados para el BoPP. Asimismo, en las líneas 132 a 139 se indica que "en la versión básica de VND, se distinguen tres fases diferentes: ...". Esta descripción corresponde a la versión básica de VNS.

3) Corregir "cosntrucciones" en la línea 144.

4) En la línea 5 del Algoritmo 1 (Sección 2.2),  ¿debe aparecer k_{max}? Este parámetro se usa normalmente para la fase de shaking del VNS y no para el VND. Además, no ha sido mencionado en el texto.

5) Con respecto a los resultados computacionales, puesto que se dispone del código correspondiente a la propuesta del estado del arte, ¿qué pasaría si se le dejara más tiempo de ejecución a dicho algoritmo, comparable con los obtenidos por el MS-VND? ¿Se alcanzaría la misma calidad?

Propongo "Strong Accept" para este trabajo.

---

### Official Review · Reviewer_aNfG · 2025-03-17
**Propuesta de algoritmo para resolver el problema BoPP, con resultados prometedores.**

**Rating:** 5
**Confidence:** 4

**Review:**

Este artículo presenta un algoritmo para la resolución del Board Packing Problem (BoPP). La propuesta consta de una fase de construcción de la solución, en la que se tienen en cuenta criterios greedy y semi-greedy, seguida de una fase de mejora mediante la aplicación de Variable Neighborhood Descent (VND). Los resultados obtenidos evidencian que el algoritmo propuesto supera en desempeño al considerado como estado del arte.

En términos generales, el manuscrito está bien redactado y presenta el contenido de manera clara y estructurada. Se valora positivamente la motivación detallada del problema realizada por los autores.

A continuación, se plantean algunas observaciones y sugerencias de mejora:

1. Los autores mencionan que el problema BoPP es NP-completo. ¿Se refieren en realidad a que es NP-hard? Sería recomendable precisar esta clasificación.

2. Las soluciones S del problema no están formalmente definidas, al igual que el conjunto de soluciones factibles SS. La función objetivo se expresa en función de las variables de decisión x_{ij} e y_r, aunque las soluciones parecen estar determinadas por las coordenadas en las que se coloca cada rectángulo (esto último no se menciona hasta el ejemplo de la Figura 2). Se deben definir formalmente las soluciones y establecer su vínculo con las variables x_{ij} e y_r, que posteriormente se utilizan para evaluar la función objetivo.

3. Es importante señalar que diferentes soluciones podrían generar los mismos valores de x_{ij} e y_r. Los autores deberían incluir una reflexión al respecto.

4. En el ejemplo de la Figura 2, en la descripción de la Figura 2b y la solución S_2, en realidad se hace referencia a la Figura 2c y la solución S_3. De igual manera, cuando se menciona la Figura 2c y la solución S_3, se está haciendo referencia a la Figura 2b y la solución S_2.

5. En la línea 144 (página 4), corregir “construcciones”.

6. En la Sección 2.2, es necesario explicar qué representan P_1(r) y P_2(r).

7. En la fase constructiva, se decide qué rectángulos se colocan y en qué posición. Esta fase finaliza cuando la colocación de cualquier rectángulo adicional supone un empeoramiento del valor de la función objetivo. Posteriormente, en la fase VND, se optimiza la distribución de los rectángulos previamente colocados. Sin embargo, surge la duda de si, una vez alcanzado un óptimo local respecto a las vecindades consideradas, sería pertinente revaluar la posibilidad de incorporar nuevos rectángulos (volviendo a la fase constructiva partiendo de esta nueva colocación). Es decir, tras la recolocación de los rectángulos, podría darse el caso de que la inclusión de nuevos elementos ya no empeorara el valor de la función objetivo. ¿Es esto correcto? Se sugiere que los autores incluyan una discusión sobre este aspecto.

---

### Decision · Program_Chairs · 2025-03-20

Accept